# Hepatitis B vaccination status and knowledge, attitude, and practice towards Hepatitis B virus among medical sciences students: A cross-sectional study

**Ibrahim A. Naqid* , Ahmed A. Mosa, Shah Vahel Ibrahim, Nizar Hussein Ibrahim, Nawfal R. Hussein**

Department of Biomedical Sciences, College of Medicine, University of Zakho, Zakho, Kurdistan Region, Iraq

* ibrahim.naqid@uoz.edu.krd

## Abstract

**Data Availability Statement:** All data is available within the paper and Supporting information file without any restriction.

### Background and aims

Healthcare staff are at high risk of occupational exposure to Hepatitis B and other blood-borne diseases. Lack of education about the knowledge of Hepatitis B virus contributes to an increase in cases. This study aims to determine the knowledge of the Hepatitis B virus among the medical professionals in Duhok province, Kurdistan Region of Iraq, and to determine their knowledge of the importance of vaccination.

### Materials and methods

This cross-sectional study was conducted in Duhok province, Kurdistan Region of Iraq, among medical science students from November 2022 to February 2023 and a total of 511 students participated in the study. A Self-administered questionnaire comprising 22 items categorized into five sections was distributed to the students either electronically or by paper and pen method. The survey utilized a Five-point Likert scale when assessing respondents' opinions on knowledge, attitude, and practice (KAP). Microsoft Excel and GraphPad Prism 9 were used for statistical analysis.

### Results

A total of 511 responses were collected from medical, dental, pharmacy, and laboratory students. The average age of the participants was 20.74 ±1.43 years. Among the respondents, only 96 (18.8%) were fully vaccinated against the Hepatitis B virus (received 3 or more doses of the vaccine), while 294 (57.5%) were not vaccinated. Lack of vaccination programs was the major reason for not receiving a vaccination (n = 182, 62%). About 286 (55.96%) of the participants had good knowledge, attitude, and practice on Hepatitis B, manifesting median scores of 26, 18, and 20, respectively.

**Funding:** The author(s) received no specific funding for this work.

**Competing interests:** The authors have declared that no competing interests exist.

## Conclusion

In our study, half of the students were found to be unvaccinated, mainly due to the absence of vaccination programs. Vaccinated students exhibited better knowledge, attitude, and practice toward the infection than non-vaccinated students. Therefore, we recommend the implementation of a vaccination program as well as training on infection prevention guidelines to increase awareness and encourage vaccination.

## Introduction

Hepatitis B virus (HBV) infection has been identified as the most common viral infection worldwide. Nearly two billion people have been infected with it, and nearly half a billion people are chronic virus carriers [1]. The 2019 Global Burden of Disease study documented 555,000 deaths worldwide attributable to diseases related to the Hepatitis B virus (HBV), and it reported a decline in the prevalence of HBV [2]. The route of transmission of HBV is through contact with infected blood or semen. There are three major modes of transmission: (1) transmission from infected mothers to neonates prenatally, (2) sexual transmission, and (3) unsafe injections, dialysis, and blood transfusions. Screening of blood donors for HBsAg has been a vital precaution in reducing the prevalence of the Hepatitis B virus [3].

Hepatitis B vaccines have been available in the United States since 1981 [4]. It is reported that with the increase of age, the antibody response declines. Immunization during childhood or adolescence offers the most potent protection against the virus. [5]. Infection with the Hepatitis B virus has been associated with predisposing patients to develop liver diseases such as liver cirrhosis, liver failure, and hepatocellular carcinoma. The Hepatitis B vaccine has been effective in the prevention of hepatocellular carcinoma [6].

Healthcare staffs are at high risk of occupational exposure to Hepatitis B, Hepatitis C, and other blood-borne diseases. In developing countries, the poor emphasis on awareness and knowledge of blood-borne diseases puts the population at higher risk [7]. In 2017, a study in Duhok, Iraq, focused on and interpreted the knowledge of Hepatitis B among healthy volunteers. In that study, it was reported that more than 40% of the participants did not know that the vaccine was available in Duhok City, and it was suggested that more awareness should be promoted in the region [8].

In 2016, a study was conducted in Duhok, Kurdistan Region, Iraq, to view the prevalence of Hepatitis B and C viruses among blood donors in the region [9]. According to the study, it was stated that none of the donors obtained vaccination against HBV due to the initiation of vaccination programs began in 2003, which meant that only people younger than 12 years old were vaccinated. Secondly, the unavailability was a serious barrier for not inoculation in the region. Lastly, the lack of awareness of the vaccine had a huge role in poor vaccination status among the donors. Lack of education about the knowledge of Hepatitis B virus aids in increasing the disease. There are also high-risk factors in Duhok City, including drug users sharing contaminated needles, sexual encounters with infected people, and healthcare staff [10, 11].

### Aims of the study

Our study aims to determine the knowledge, attitude and practice (KAP) towards the Hepatitis B virus among the medical science students in Duhok, Kurdistan Region of Iraq. Additionally, the study aims to determine the vaccination status of these students, exploring the underlying reasons for those who have not been vaccinated. This will help in encouraging awareness and

educating medical sciences staff in particular and the population in general to protect themselves from the virus, and advocate for the effective role of the vaccine against the disease initiation and progression.

## Materials and methods

### Study design

This cross-sectional study was carried out among medical science students in Duhok province, Kurdistan Region of Iraq. The data were collected from November 2022 to February 2023, and a total of 511 students were recruited for the study. A self-administered questionnaire was delivered to students electronically using both the Google Forms platform and the paper and pen method. The research was conceptualized and conducted according to the Standards for Strengthening the Reporting of Observational Studies in Epidemiology.

### Study tools

The study survey was derived from a previously validated questionnaire with certain adjustments made by the researchers to meet the current study aims [12]. The questionnaire comprised 22 items which were categorized into five sections. The first section included three questions related to the basic demographic characteristics of participants like age, gender, and college. The second section featured three items regarding the vaccination status, Hepatitis B vaccine doses, and the reason for not being vaccinated.

The last part consisted of questions designed to examine students' knowledge, attitude, and practice on Hepatitis B, which may be useful in understanding students' acceptance and reticence regarding the Hepatitis B vaccine. This part of the questionnaire was assorted into three sections: six items to test participants' Hepatitis B knowledge, five items to assess participants' attitudes regarding the virus, and the final five questions to address students' practice.

Responses to the questions about knowledge, attitude, and practice were measured using a Five-point Likert scale, ranging from 1–5 from strongly disagree to strongly agree. However, one question in the knowledge section (i.e., Hepatitis B can be transmitted by shaking hands, coughing/sneezing, and contaminated food/water) and two questions in the attitude section (I feel uncomfortable sitting with a Hepatitis B infected person, and I don't need Hepatitis B vaccination because I am not at risk) were reversed scored. Knowledge of Hepatitis B was examined by adding six questions with a potential score ranging from 6–30. The Attitude and Practice sections were assessed by adding five questions with a possible score ranging from 5–25. A possible KAP score can vary between 16 and 80. Participants with a score at or above the median were categorized as having a favorable KAP score, while those with a score below the median were classified as having an inadequate KAP score [12].

### Inclusion/Exclusion criteria

The inclusion criteria were participants of more than 18 years of age, students in one of the medical sciences colleges in Duhok province, and consenting to be recruited in the study. At the same time, non-medical science students from other provinces, as well as data with incomplete information, were excluded from the study.

### Statistical analysis

Statistical analysis was performed using Microsoft Excel and GraphPad Prism 9 software. Frequencies and percentages were used to describe descriptive statistics of the participants. The association between basic-demographic characteristics variables and KAP score was assessed

by using Chi-Square, or Fisher's exact test. Statistical significance was defined as a p-value of 0.05 or less.

## Results

### Basic demographic characteristics

The basic demographic characteristics of the respondents who participated in this study are shown in Table 1. The mean age of the participants was 20.74 (±1.43 SD) years. Two hundred seventy-one respondents were female (53%). The majority of the respondents were medical students (n = 248, 48.5%) followed by dental students (n = 161, 31.5%). Regarding the vaccination status, 217 (42.5%) of the students were vaccinated while 294 (57.5%) were not vaccinated. Among the vaccinated students, 96 (44.2%) were fully vaccinated (received 3 or more doses of the vaccine). The lack of a vaccination program was the major reason for not being vaccinated (n = 182, 62%).

### Assessment of knowledge related to Hepatitis B virus

Table 2 shows the knowledge of the participants on Hepatitis B. Most of the respondents agreed that Hepatitis B is caused by a virus (67.5% strongly agreed, 24.1% agreed) and can

**Table 1. Demographic characteristics and Hepatitis B vaccination status of the participants.**

| Variables | n (%) |
|---|---|
| **Age** | |
| 20 and below | 243 (47.6) |
| 21 and above | 268 (52.4) |
| Mean (SD) | 20.74 (1.43) |
| **Gender** | |
| Male | 240 (47) |
| Female | 271 (53) |
| **Field of study** | |
| College of Medicine | 248 (48.5) |
| College of Dentistry | 161 (31.5) |
| College of Pharmacy | 52 (10.2) |
| Medical Laboratory College | 50 (9.8) |
| **Vaccinated against Hepatitis B** | |
| Yes | 217 (42.5) |
| No | 294 (57.5) |
| **Doses of Hepatitis B vaccine received** | |
| Not vaccinated | 294 (57.5) |
| One | 41 (8) |
| Two | 80 (15.7) |
| Three | 87 (17) |
| More than three | 9 (1.8) |
| **Reasons for not being vaccinated against Hepatitis B** | |
| No vaccination program offered | 182 (62) |
| Low risk of Hepatitis B | 26 (8.8) |
| Not sure about the vaccination status | 22 (7.5) |
| Lack of knowledge | 36 (12.2) |
| Efficacy doubted | 28 (9.5) |

**Table 2. Assessment of knowledge related to Hepatitis B virus.**

| Variables | Strongly Disagree n (%) | Disagree n (%) | Neutral n (%) | Agree n (%) | Strongly Agree n (%) |
|---|---|---|---|---|---|
| 1. Hepatitis B is caused by a virus. | 10 (2) | 8 (1.6) | 25 (4.9) | 123 (24.1) | 345 (67.5) |
| 2. Hepatitis B can be transmitted by contaminated blood, body fluids, and unprotected sex. | 11 (2.2) | 22 (4.3) | 30 (5.9) | 130 (25.4) | 318 (62.2) |
| 3. Hepatitis B can be transmitted by shaking hands, Coughing/Sneezing, and contaminated food/water. | 249 (48.7) | 139 (27.2) | 56 (11) | 38 (7.4) | 29 (5.7) |
| 4. Hepatitis B can cause liver cancer. | 13 (2.5) | 23 (4.5) | 96 (18.8) | 197 (38.6) | 182 (35.6) |
| 5. Healthcare workers are at increased risk of getting Hepatitis B than the general population. | 14 (2.7) | 26 (5.1) | 67 (13.1) | 144 (28.2) | 260 (50.9) |
| 6. Hepatitis B can be prevented by vaccination, using gloves and avoiding sharp needles/syringe injury. | 9 (1.8) | 10 (2) | 32 (6.3) | 175 (34.2) | 285 (55.8) |

cause liver cancer (35.6% strongly agreed, 38.6% agreed). Regarding the knowledge of the participants on the mode of transmission, the majority of the students agreed that Hepatitis B can be transmitted by contaminated blood, body fluids, and unprotected sex (62.2% strongly agreed, 25.4% agreed). Similarly, most of the students disagreed that Hepatitis B can be transmitted by shaking hands, coughing/sneezing, and contaminated food/water (48.7% strongly disagreed, 27.2% disagreed) and 11% were neutral. About four-fifths of the participants agreed that healthcare workers are at increased risk of contracting Hepatitis B than the general population (50.9% strongly agreed, 28.2% agreed). In terms of knowledge on prevention, the majority of the students agreed that Hepatitis B can be prevented by vaccination, using gloves, and avoiding sharp needles/syringe injury (55.8% strongly agreed, 34.2% agreed).

## Assessment of attitude towards Hepatitis B virus

Table 3 shows the attitude of the participants toward Hepatitis B. Many of the students were neutral (n = 140, 27.4%) toward sitting with a Hepatitis B-infected person, however, 14.3% (n = 73) of the participants strongly agreed on feeling uncomfortable sitting with a Hepatitis B infected person and 14.5% (n = 74) strongly disagreed. Twenty-seven percent (n = 138) disagreed on shaking hands/hugging with an infected person, while 21.3% (n = 109) did not mind shaking hands/hugging with a Hepatitis B infected person and 25.4% (n = 130) were neutral. About 41.1% (n = 210) of students agreed that the Hepatitis B vaccine is safe and effective and 3.9% (n = 20) strongly disagreed. About two-thirds of the participants (66.7%, n = 341) strongly agreed that healthcare workers should receive Hepatitis B vaccination. About half of the respondents (52.6%, n = 269) strongly disagreed with not receiving Hepatitis B vaccination because they were not at risk.

**Table 3. Assessment of attitude towards Hepatitis B virus.**

| Variables | Strongly Disagree n (%) | Disagree n (%) | Neutral n (%) | Agree n (%) | Strongly Agree n (%) |
|---|---|---|---|---|---|
| 1. I feel uncomfortable sitting with a Hepatitis B infected person. | 74 (14.5) | 93 (18.2) | 140 (27.4) | 131 (25.6) | 73 (14.3) |
| 2. I don't mind shaking hands/hugging with a Hepatitis B infected person. | 77 (15.1) | 138 (27) | 130 (25.4) | 109 (21.3) | 57 (11.2) |
| 3. I believe the Hepatitis B vaccine is safe and effective. | 20 (3.9) | 19 (3.7) | 76 (14.9) | 210 (41.1) | 186 (36.4) |
| 4. I believe healthcare workers should receive Hepatitis B vaccination. | 6 (1.2) | 16 (3.1) | 37 (7.2) | 111 (21.7) | 341 (66.7) |
| 5. I don't need Hepatitis B vaccination because I'm not at risk. | 269 (52.6) | 142 (27.8) | 45 (8.8) | 30 (5.9) | 25 (4.9) |

**Table 4. Assessment of practice towards Hepatitis B virus.**

| Variables | Strongly Disagree n (%) | Disagree n (%) | Neutral n (%) | Agree n (%) | Strongly Agree n (%) |
|---|---|---|---|---|---|
| 1. I ask/use a new blade for shaving/hair cutting. | 26 (5.1) | 43 (8.4) | 103 (20.2) | 144 (28.2) | 195 (38.2) |
| 2. I ask for a new syringe before injection. | 8 (1.6) | 23 (4.5) | 48 (9.4) | 123 (24.1) | 309 (60.5) |
| 3. I ask for sterilized equipment for ear/nose piercing. | 8 (1.6) | 24 (4.7) | 87 (17) | 135 (26.4) | 257 (50.3) |
| 4. I will report for needle prick/ sharp injuries. | 4 (0.8) | 32 (6.3) | 107 (20.9) | 151 (29.5) | 217 (42.5) |
| 5. I attend Hepatitis B-related awareness campaigns. | 71 (13.9) | 68 (13.3) | 141 (27.6) | 139 (27.2) | 92 (18) |

## Assessment of practice towards Hepatitis B virus

Table 4 shows the practice of the respondents toward Hepatitis B. Among the participants, 38.2% (n = 195) asked/used a new blade for shaving/hair cutting. About sixty percent (n = 319) of the respondents asked for a new syringe before injection. About half of the students (50.3%, n = 257) ask for sterilized equipment for ear/nose piercing. While 42.5% (n = 217) stated that they would report for needle prick/ sharp injuries. Approximately one-third of students (n = 141) were neutral on attending Hepatitis B-related awareness campaigns and 13.9% (n = 71) strongly disagreed.

## Categorization of knowledge, attitude, and practice (KAP) score and its association with basic demographic characteristics and HBV vaccination status

The total median score of KAP was 65 as shown in Table 5. The median knowledge, attitude and practice score was 26,19,20, respectively Table 5. Table 6 shows the association between KAP score and basic demographic characteristics and HBV vaccination status including age, gender, the field of study (college), vaccination against the virus, doses of HBV vaccine, and reasons for not being vaccinated against HBV. We found a significant association between the KAP score and most of the variables. However, there was no significant association between gender (p-value = 0.093) and KAP score Table 6.

## Discussion

Medical science students are at high risk of exposure to blood-borne infections such as Hepatitis B virus, as they provide direct patient care throughout the clerkship phase of the program, this places them in a similar risk category as healthcare professionals [13]. Therefore, it is imperative that all students should be vaccinated and acquire sufficient knowledge about the virus to minimize the risk of infection. In the Kurdistan Region of Iraq, there have been no studies to measure vaccination status as well as knowledge, attitude, and practice towards the Hepatitis B virus among medical sciences students, as far as authors' knowledge are concerned. This study is designed to evaluate the vaccination status of medical sciences students and their KAP about the virus.

**Table 5. Summation of knowledge, attitude, and practice score distribution.**

| | Knowledge sum (n = 511) | Attitude sum (n = 511) | Practice sum (n = 511) | Total Score (n = 511) |
|---|---|---|---|---|
| Mean | 25.61 | 18.5 | 19.7 | 63.81 |
| Median | 26 | 19 | 20 | 65 |

**Table 6. Association of KAP score with basic-demographic characteristics.**

| Variables | | KAP score Category | | *p-value |
|---|---|---|---|---|
| | | In-adequate (<65)n (%) | Good (≥65) n (%) | |
| **Age** | 20 and below (n = 243) | 142 (58.4) | 101 (41.6) | 0.00002 |
| | 21 and above (n = 268) | 106 (39.6) | 162 (60.4) | |
| **Gender** | Male (n = 240) | 126 (52.5) | 114 (47.5) | 0.093 |
| | Female (n = 271) | 122 (45.02) | 149 (54.98) | |
| **Field of study (College)** | Medicine (n = 248) | 112 (45.16) | 136 (54.84) | 0.0002 |
| | Dentistry (n = 161) | 73 (45.34) | 88 (54.66) | |
| | Pharmacy (n = 52) | 22 (42.31) | 30 (57.69) | |
| | Medical Laboratory (n = 50) | 41 (82) | 9 (18) | |
| **Vaccinated against Hepatitis B** | Yes (n = 217) | 85 (39.17) | 132 (60.83) | 0.0003 |
| | No (n = 294) | 163 (55.44) | 131 (44.56) | |
| **Doses of Hepatitis B vaccine received** | Not vaccinated (n = 294) | 163 (55.4) | 131 (44.56) | 0.000428 |
| | One (n = 41) | 19 (46.3) | 22 (53.7) | |
| | Two (n = 80) | 38 (47.5) | 42 (52.5) | |
| | Three (n = 87) | 26 (29.9) | 61 (70.1) | |
| | More than three (n = 9) | 2 (22.2) | 7 (77.8) | |
| **Reasons for not being vaccinated against Hepatitis B** | No vaccination program offered (n = 182) | 85 (46.7) | 97 (53.3) | 0.000851 |
| | Low risk of Hepatitis B (n = 26) | 19 (73.1) | 7 (26.9) | |
| | Not sure about the vaccination status (n = 22) | 12 (54.5) | 10 (45.5) | |
| | Lack of knowledge (n = 36) | 24 (66.7) | 12 (33.3) | |
| | Efficacy doubted (n = 28) | 23 (82.1) | 5 (17.9) | |

*P value is measured using Chi-Square, or Fisher's exact test.

In our study, 42.5% of medical science students were vaccinated against Hepatitis B, which is higher than the findings of a study in Southwest Ethiopia, where only 25.7% of students were immunized [14]. However, our results are similar to a study from Pakistan, where 42.2% of students were vaccinated, and are quite lower than a study from Nigeria, where 47.7% of students reported that they had been vaccinated [15, 16]. Also, it is lower than research carried out in Uganda, where it was found that 66.8% of the participants had received vaccinations [17]. On the other hand, a more current study conducted in Pakistan found that 79% of their participants are vaccinated against Hepatitis B [18]. All in all, 18.8% of participants were fully vaccinated (3 doses or more), which is much greater than a study in Ethiopia, where just 2% of students were fully vaccinated [19]. In comparison to our findings, research in Nepal found that 37% of their students had completed all three doses of vaccination, which is greater than our study [12].

The primary reason for why 57.5% of the participants in our study were not vaccinated is due to having no vaccination programs (62%) followed by lack of knowledge (12.2%) and efficacy doubted (9.5%). Our results are in line with other studies conducted in Nepal and Nigeria, which found a lack of effective vaccination programs (43.2%) and a lack of opportunity (57.4%) as the main reasons for non-vaccination, respectively [16, 20]. These findings highlight the need for urgent implementation of vaccination programs for medical science students in the region. In Uganda, the high cost of vaccination (63.2%) was the most common obstacle for non-vaccination [17].

About 91.6% of the participants surveyed were aware that Hepatitis B is caused by a virus. Compared to a research study carried out in Nepal among medical and dental students, it was

found that 93.6% of the students knew that Hepatitis B infection was caused by a virus [21]. The great majority of participants have a solid understanding of the mode of transmission, with 87.6% stating that transmission occurs through contaminated blood, fluid, and unprotected intercourse. These findings are consistent with those of Ethiopian and Nepalese studies [12, 19]. In the present study, 74.2% of participants agreed that the Hepatitis B virus can lead to liver cancer. This is comparable with research studies from Saudi Arabia, Nepal, and Ethiopia which revealed that 75.5%, 80.6%, and 81.3% of students agreed that Hepatitis B infection can cause liver cancer, respectively [12, 19, 22]. Ninety percent of respondents agreed that vaccination may prevent Hepatitis B infection. In this regard, comparable results were observed in studies carried out in Saudi Arabia, Nepal, and Ethiopia [12, 19, 22].

Around 39.9% and 42.1% of students showed a negative attitude towards sitting and shaking hands with a Hepatitis B infected person, respectively. Nepalese students demonstrated a more positive attitude [12]. In the present study, 77.5% of the students believed that the Hepatitis B vaccine was safe and efficient; somewhat higher results were obtained from studies conducted in Saudi Arabia, Nepal, and Ethiopia [12, 19, 22]. While compared to another study in Saudi Arabia, our findings are higher where only 63% of participants agreed that the vaccine is safe and effective [23]. A high enough percentage (88.4%) of our students agreed that healthcare workers should be vaccinated, which is similar to a study from Nepal [12]. According to our survey, 10.8% of participants believed that they were not at risk of contracting the Hepatitis B virus and that vaccination against Hepatitis B is not necessary. This figure nearly triples when compared to studies conducted in India and Nepal, whereas only 3.7% and 3.9% of their participants agreed with this statement, respectively [12, 24]. This concerned finding should be addressed because medical science students are part of the healthcare system and should be aware of the need for Hepatitis B vaccination since they are vulnerable to contracting the virus and spreading the infection in the community.

In our survey, 66.4%, 84.6%, and 76.7% of the students stated that they requested a new blade for shaving/hair cutting, new syringes to avoid infection, and sanitized equipment for piercing their ears and nose, respectively. Studies from India and Nepal reported better safety practices among their participants [12, 24]. These findings highlight the necessity of taking action that encourages students to adhere to infection control measures. 72% of the students agreed that they would report needle prick/sharp injuries. Our findings presented higher percentages than those indicated by studies in Saudi Arabia (68%), Nepal (64.6%), and Ethiopia (53.7%), where the participants will report needle injuries [12, 19, 22]. Also, similar to the findings of a study in Nepal (i.e., 44.2%), a percentage of 45.2% of the study participants said that they have attended Hepatitis B related awareness programs [12].

In this study, we found that the female gender is associated with better overall KAP scores when compared to males. These findings are similarly reported in the previous studies from Pakistan and Nepal [12, 18]. However, a study from Malaysia revealed no association between gender and knowledge about Hepatitis B [25]. Students who were vaccinated showed a better KAP score than those who were not. A similar result was reported by a Nepalese study [12].

## Limitations and strengths of the study

The main limitation of this research is that vaccination status was self-reported and not confirmed by the measurement of the anti-Hepatitis B surface antibody (HBsAb) titer of the students. Therefore, recall bias and erroneous information might have affected the findings of the research. Since the study was conducted only in Duhok province in the Kurdistan Region of Iraq, the results cannot be generalized to all medical science colleges in the region, but they

will undoubtedly serve as a background for future studies in the region. Finally, the nature of the study was cross-sectional, and it did not measure the cause-and-effect relationship. Despite the limitations, the study was strengthened by the large sample size used to measure individuals' vaccination status and KAP against the Hepatitis B virus, which minimized the likelihood of bias.

## Conclusions

In the present study, more than half of the participants were not vaccinated against Hepatitis B despite it being a preventable disease, and only one-fifth of students were fully vaccinated. The lack of vaccination programs was the main reason for the study participants not to be vaccinated against Hepatitis B. Moreover, about half of the participants had good knowledge, attitude, and practice (KAP) regarding Hepatitis B, and students who were vaccinated displayed a better KAP score than those who were non-vaccinated. The administrators of the medical sciences college should prioritize the implementation of a vaccination program.

## Recommendations

Since all medical science students are susceptible to Hepatitis B infection as they are exposed to the infected blood and blood products in their professional career, therefore, all students should be vaccinated against the Hepatitis B virus upon enrollment at the college. The COVID-19 pandemic has had a significant impact on the region's healthcare system including medical education [26–28]. This has negatively impacted and postponed many infection prevention programs in the Kurdistan Region [29]. This might be linked to low vaccination coverage in our study. According to Khan et al., a pretest, followed by a lecture and demonstration of standard infection control measures and precautions, and concluded with a posttest, has the potential to enhance knowledge and bring about attitudinal changes [30]. Therefore, we recommend the implementation of a vaccination program and the provision of training on infection prevention guidelines through lectures and workshops to increase students' knowledge, attitude, and practice toward Hepatitis B.

## Supporting information

**S1 Dataset.**
(XLSX)

## Acknowledgments

We would like to express sincere gratitude to all students who took part in this study.

## Author Contributions

**Conceptualization:** Ibrahim A. Naqid.

**Data curation:** Ahmed A. Mosa, Shah Vahel Ibrahim, Nizar Hussein Ibrahim.

**Formal analysis:** Ibrahim A. Naqid, Ahmed A. Mosa, Shah Vahel Ibrahim, Nizar Hussein Ibrahim.

**Methodology:** Ahmed A. Mosa, Shah Vahel Ibrahim, Nizar Hussein Ibrahim.

**Supervision:** Ibrahim A. Naqid.

**Validation:** Nawfal R. Hussein.

**Visualization:** Ibrahim A. Naqid, Nawfal R. Hussein.

**Writing – original draft:** Ahmed A. Mosa, Shah Vahel Ibrahim.

**Writing – review & editing:** Ibrahim A. Naqid, Nizar Hussein Ibrahim, Nawfal R. Hussein.

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
