## [Decision Letter · Decision Letter 0]

6 Sep 2023

PONE-D-23-22175Hepatitis B Vaccination Status and Knowledge, Attitude, and Practice towards Hepatitis B virus among Medical Sciences Students: A Cross-Sectional StudyPLOS ONE

Dear Dr. Naqid,

Thank you for submitting your manuscript to PLOS ONE. After careful consideration, we feel that it has merit but does not fully meet PLOS ONE’s publication criteria as it currently stands. Therefore, we invite you to submit a revised version of the manuscript that addresses the points raised during the review process.

ACADEMIC EDITOR:- The manuscript needs an extensive proof-readingfor English language.

- In the abstract, you should provide what was the scale used. Giving just a score of 20 or 26 has no sense.

- Improve the quality of your tables.

- You should provide the references used to categorize the scores. In fact, considering those who have a score of 64 with those having an evantual score  of 10 is ambigous. 

- Figure 1 is not necessary.

- Table 1 may be "demographics and HBV vaccination status. 

 - In table 6 you can include the other factors that could be associated with tKAP.mAP. ==============================

We look forward to receiving your revised manuscript.

Kind regards,

Mohamed Lounis

Academic Editor

PLOS ONE

Reviewers' comments:

Reviewer's Responses to Questions

**Comments to the Author**

1. Is the manuscript technically sound, and do the data support the conclusions?

Reviewer #1: Yes

Reviewer #2: Yes

2. Has the statistical analysis been performed appropriately and rigorously? 

Reviewer #1: Yes

Reviewer #2: Yes

3. Have the authors made all data underlying the findings in their manuscript fully available?

Reviewer #1: Yes

Reviewer #2: Yes

4. Is the manuscript presented in an intelligible fashion and written in standard English?

Reviewer #1: No

Reviewer #2: Yes

5. Review Comments to the Author

Reviewer #1: The manuscript needs to be revised by a native English speaker as there are many typographical errors and grammatical mistakes.

Introduction is good but need to improve the quality of writing (English proofreading).

Why is 2010 Global Burden of Disease study being quoted when there are far more recent reports from the GBD.

Too much detail about mode of transmission in first paragraph. Can be reduced. The main mode of transmission to be concerned with is that of needle stick injuries…etc. The rest can be summarized.

Methodology is sound but give information on if the study was piloted and what language the questionnaire was in. Reference 12 (Nepal study) that the study was based on was in English I presume, so if there was a translated questionnaire, was anything done to ensure validity and reliability of the translated questionnaire?

Discussion focuses too much on comparison with the Nepal study and a Saudi one. Was a thorough literature search conducted to gain more of a worldwide view on the matter?

The study did not explore whether or not students would get vaccinated if it were a prerequisite by their college for clinical training, as that would be a strong recommendation if the results of this question yielded a high response.

Reviewer #2: Revise the manuscript according to the suggestions and recommendations as indicated in the revised manuscript. Many grammatical mistakes. It is better to show the article to a English consultant to improve the language.

6. PLOS authors have the option to publish the peer review history of their article (what does this mean?). If published, this will include your full peer review and any attached files.

Reviewer #1: No

Reviewer #2: **Yes: **Dr. Nazeer Khan

---

## [Author Response · Author response to Decision Letter 0]

18 Sep 2023

The Response to Reviewers Letter

PLOS ONE JOURNAL 

Dear Editor 

Thank you for your kind comments, we found the editorial board and reviewers' comments on our submitted manuscript draft to be very valuable. We believe that our paper is significantly strengthened after responding to the comments.

We have carefully reviewed the comments and have modified the manuscript accordingly. The revised points are highlighted in RED in the revised manuscript file. 

Response to “ACADEMIC EDITOR” comments 

1. The manuscript needs extensive proof-reading for English language 

- The manuscript has been revised for typo-grammatical errors by an English language editor 

2. In the abstract, you should provide what was the scale used. Giving just a score of 20 or 26 has no sense

- Five-point Likert scale has been added to the abstract in the materials and methods section. 

3. Improve the quality of your tables.

- Done

4. You should provide the references used to categorize the scores. In fact, considering those who have a score of 64 with those having an eventual score of 10 is ambiguous.

-Addressed as appropriate in Materials and Methods section 

5. Figure 1 is not necessary.

-Removed 

6. Table 1 may be "demographics and HBV vaccination status. 

-Done 

7. In table 6 you can include the other factors that could be associated with KAP

-Done 

All comments provided by the editorial board are adequately responded to.

Response to “REVIEWERS” Comments 

• Reviewer #1

1. The manuscript needs to be revised by a native English speaker as there are many typographical errors and grammatical mistakes. 

- The manuscript has been revised for typo-grammatical errors by an English language editor 

2. Comments on introduction 

-All comments have been addressed 

3. Why is 2010 Global Burden of Disease study being quoted when there are far more recent reports from the GBD.

-Reference has been updated 

4. Methodology is sound but give information on if the study was piloted and what language the questionnaire was in. Reference 12 (Nepal study) that the study was based on was in English I presume, so if there was a translated questionnaire, was anything done to ensure validity and reliability of the translated questionnaire?

-Medical sciences colleges in the Kurdistan region of Iraq are in English language. So, all the students are able to answer the questionnaire in English language. Therefore, it did not require translation of the questionnaire. 

5. Discussion focuses too much on comparison with the Nepal study and a Saudi one. Was a thorough literature search conducted to gain more of a worldwide view on the matter.

– The study was compared to neighboring countries and other developing countries.

• Reviewer #2

-All the comments provided by reviewer #2 in the reviewed Microsoft Word file are addressed as appropriate. 

Modification to References 

- Reference number [2] has been updated as appropriate.

- Reference number [12] has been added to last sentence of Material and Methods section/Study tool subheading. 

Note: All data are available within the manuscript and supporting information without any restriction 

We hope that the manuscript is now suitable for your journal, looking forward to hearing from you.

Sincerely,

Asst. Prof. Dr Ibrahim A. Naqid

Department of Biomedical Sciences, 

College of Medicine, University of Zakho

Zakho International Road, Duhok, Kurdistan Region-Iraq

P.O. Box12

Email: ibrahim.naqid@uoz.edu.krd

Tel:. 009647504737593

---

## [Decision Letter · Decision Letter 1]

3 Oct 2023

PONE-D-23-22175R1Hepatitis B Vaccination Status and Knowledge, Attitude, and Practice towards Hepatitis B virus among Medical Sciences Students: A Cross-Sectional StudyPLOS ONE

Dear Dr. Naqid,

Thank you for submitting your manuscript to PLOS ONE. After careful consideration, we feel that it has merit but does not fully meet PLOS ONE’s publication criteria as it currently stands. Therefore, we invite you to submit a revised version of the manuscript that addresses the points raised during the review process.

ACADEMIC EDITOR:You should take into consideration all the comments of the Editor ad the reviewers.You should take into consideration the comments of the atached fie.==============================

We look forward to receiving your revised manuscript.

Kind regards,

Mohamed Lounis

Academic Editor

PLOS ONE

Journal Requirements:

Reviewers' comments:

Reviewer's Responses to Questions

**Comments to the Author**

1. If the authors have adequately addressed your comments raised in a previous round of review and you feel that this manuscript is now acceptable for publication, you may indicate that here to bypass the “Comments to the Author” section, enter your conflict of interest statement in the “Confidential to Editor” section, and submit your "Accept" recommendation.

Reviewer #2: (No Response)

2. Is the manuscript technically sound, and do the data support the conclusions?

Reviewer #2: (No Response)

3. Has the statistical analysis been performed appropriately and rigorously? 

Reviewer #2: Yes

4. Have the authors made all data underlying the findings in their manuscript fully available?

Reviewer #2: Yes

5. Is the manuscript presented in an intelligible fashion and written in standard English?

Reviewer #2: Yes

6. Review Comments to the Author

Reviewer #2: Author has addressed most of the comments, but did not address few of them, and did not write why those comments are unattended. Please see my previous comments in the attachemnt.

7. PLOS authors have the option to publish the peer review history of their article (what does this mean?). If published, this will include your full peer review and any attached files.

Reviewer #2: **Yes: **Dr. Nazeer Khan

---

## [Author Response · Author response to Decision Letter 1]

11 Oct 2023

The Response to Reviewers Letter

PLOS ONE JOURNAL 

Dear Editor 

Thank you for your kind comments, we found the remaining unaddressed reviewer’s comments on our submitted manuscript draft to be very valuable. We believe that our paper is further strengthened after responding to the remaining comments.

We have carefully reviewed the comments and have modified the manuscript accordingly. The revised points are highlighted in BLUE in the revised manuscript file. 

Response to remaining “REVIEWERS” Comments 

• Reviewer #2

1. Materials and Methodology: Year of study of the students?

- Unfortunately, has not been included in our questionnaire. 

2. Results: Dividing the favorable and non-favorable at median?

-It is cited in the methodology that we followed reference 12. 

3. Discussion: Reference that supports the below statement “Medical students are at comparable risk of getting hepatitis B compared to Health care workers” 

-A new reference has been added.

4. Discussion: Comparing to other Pakistan study, reference 23 in the old file and reference 18 in the new file 

-Done.

5. Discussion: Does Nepali private hospitals offer vaccination services with some cost. Then cost could be one of the barriers for not vaccination.

-Private hospitals in Nepal provide Hepatitis B vaccination with some cost, but according to what’s available in the literature, we found that cost is not a major barrier to non-vaccination among Nepalese students. For example, in reference 12 only 2 students (1.1%) stated high cost of vaccination as a barrier.

6. A study has been added to support our recommendations 

Modification to References 

- Two references have been added newly, reference number [13] and [30]. 

- Reference number [23] has been changed to reference number [18].

We hope that the manuscript is now suitable for your journal, looking forward to hearing from you.

Sincerely,

Asst. Prof. Dr Ibrahim A. Naqid

Department of Biomedical Sciences, 

College of Medicine, University of Zakho

Zakho International Road, Duhok, Kurdistan Region-Iraq

P.O. Box12

Email: Ibrahim.naqid@uoz.edu.krd

Tel:. 009647504737593

---

## [Decision Letter · Decision Letter 2]

18 Oct 2023

PONE-D-23-22175R2Hepatitis B Vaccination Status and Knowledge, Attitude, and Practice towards Hepatitis B virus among Medical Sciences Students: A Cross-Sectional StudyPLOS ONE

Dear Dr. Naqid,

Thank you for submitting your manuscript to PLOS ONE. After careful consideration, we feel that it has merit but does not fully meet PLOS ONE’s publication criteria as it currently stands. Therefore, we invite you to submit a revised version of the manuscript that addresses the points raised during the review process.

We look forward to receiving your revised manuscript.

Kind regards,

Mohamed Lounis

Academic Editor

PLOS ONE

Additional Editor Comments:

In your tables Change "No." to "n".

Reviewers' comments:

Reviewer's Responses to Questions

**Comments to the Author**

1. If the authors have adequately addressed your comments raised in a previous round of review and you feel that this manuscript is now acceptable for publication, you may indicate that here to bypass the “Comments to the Author” section, enter your conflict of interest statement in the “Confidential to Editor” section, and submit your "Accept" recommendation.

Reviewer #2: All comments have been addressed

2. Is the manuscript technically sound, and do the data support the conclusions?

Reviewer #2: Yes

3. Has the statistical analysis been performed appropriately and rigorously? 

Reviewer #2: Yes

4. Have the authors made all data underlying the findings in their manuscript fully available?

Reviewer #2: Yes

5. Is the manuscript presented in an intelligible fashion and written in standard English?

Reviewer #2: Yes

6. Review Comments to the Author

Reviewer #2: You revised the manuscript according to the suggestions. However, few more minor ones were left without giving any justification.

7. PLOS authors have the option to publish the peer review history of their article (what does this mean?). If published, this will include your full peer review and any attached files.

Reviewer #2: **Yes: **Dr. Nazeer Khan

---

## [Author Response · Author response to Decision Letter 2]

18 Oct 2023

Thanks for your comment, In all tables No. changed into “n” in green color

---

## [Editor Report · Decision Letter 3]

20 Oct 2023

Hepatitis B Vaccination Status and Knowledge, Attitude, and Practice towards Hepatitis B virus among Medical Sciences Students: A Cross-Sectional Study

PONE-D-23-22175R3

Dear Dr. Naqid,

We’re pleased to inform you that your manuscript has been judged scientifically suitable for publication and will be formally accepted for publication once it meets all outstanding technical requirements.

Kind regards,

Mohamed Lounis

Academic Editor

PLOS ONE
---

## [Editor Report · Acceptance letter]

27 Oct 2023

PONE-D-23-22175R3 

Hepatitis B Vaccination Status and Knowledge, Attitude, and Practice towards Hepatitis B Virus among Medical Sciences Students: A Cross-Sectional Study 

Dear Dr. Naqid:

I'm pleased to inform you that your manuscript has been deemed suitable for publication in PLOS ONE. Congratulations! Your manuscript is now with our production department. 

Kind regards, 

on behalf of

Dr. Mohamed Lounis 

Academic Editor

PLOS ONE